# COVID-19 vaccine acceptance and hesitancy in N'Djamena, Chad: A cross-sectional study of patients, community members, and healthcare workers

**Dylan R. Rice**[1], **Anatole Balamo**[2], **Allah-Rabaye Thierry**[2], **Aremadji Gueral**[2], **Djerakoula Fidele**[2], **Farrah J. Mateen**[1] *, **Foksouna Sakadi**[2]

**1** Department of Neurology, Massachusetts General Hospital, Boston, Massachusetts, United States of America, **2** Neurology Unit, National Reference Teaching Hospital, N'Djamena, Chad

* fmateen@mgh.harvard.edu

**Data Availability Statement:** De-identified data are available in supplementary materials.

## Abstract

As of March 2022, the COVID-19 vaccination rate in Chad approximated 1%. There are no published reports of COVID-19 vaccine hesitancy or beliefs in Chad. We aimed to study COVID-19 vaccine acceptance and hesitancy among community members, patients, and health care workers in urban Chad. We recruited a prospective convenience sample of adult patients, community members, and healthcare workers from N'Djamena, Chad between August–October 2021. Participants completed a 15-minute, 25-question survey instrument exploring demographic, social, and clinical variables related to COVID-19 and an adapted WHO SAGE Vaccine Hesitancy Survey. Primary outcomes were vaccine acceptance and vaccine hesitancy. Regression models were fit to assess associations between Vaccine Hesitancy Scale (VHS) scores, ranging from 10 (least hesitant) to 50 (most hesitant) points, and pre-selected variables of interest. An inductive thematic analysis was used to analyze the qualitative vaccine hesitancy responses. Of 508 participants (32% female; mean age 32 years), 162 were patients, 153 were community members, and 193 were healthcare workers. COVID-19 vaccine acceptance was significantly higher among patients (67%) than community members (44%) or healthcare workers (47%), $p < .001$. The average VHS score was 29 points (patients = 27.0, community members = 28.9, healthcare workers = 29.4), and more than one-third of participants were classified as highly vaccine hesitant (score >30 points). Knowing someone who died from COVID-19, believing local healthcare workers support vaccination, trusting the government, having a higher socioeconomic status (i.e. having electricity), and reporting medical comorbidities were each associated with less vaccine hesitancy (all $p < .05$). The vaccine concerns most frequently endorsed were: vaccine side effects (48%), efficacy (38%), safety (34%), concerns about the pharmaceutical industry (27%), and lack of government trust (21%). Four main themes arose from qualitative vaccine hesitancy responses (n = 116): education, trust, clinical concerns, and misinformation and false beliefs. Overall, COVID-19 vaccine acceptance was low, including among health care workers, and reasons for vaccine hesitancy were broad. We detail the most commonly reported concerns of urban Chadians for receiving the COVID-19 vaccine; we also identify

**Funding:** Funding for OA fees was provided by the Massachusetts General Hospital Center for Global Health.

**Competing interests:** The authors have declared that no competing interests exist.

subgroups most likely to endorse vaccine hesitancy. These analyses may inform future vaccination outreach campaigns in N'Djamena.

## Introduction

There are several barriers to achieving the uptake of COVID-19 vaccines globally, including macro-level structural, political, and social issues. A self-reported factor associated with vaccine uptake is *vaccine hesitancy*, defined by the World Health Organization as the "delay in acceptance or refusal of vaccines despite availability of vaccine services," which is "complex and context specific, varying across time, place and vaccines" [1]. In a systematic review of COVID-19 vaccine hesitancy across 33 countries spanning 6 continents, vaccine acceptance rates tended to be ≥70% and ranged from 24% (Kuwait) to 97% (Ecuador) [2]. In a preprint of a review of COVID-19 vaccine hesitancy in 11 countries in Sub-Saharan Africa (SSA), vaccine acceptance ranged from 25% (Nigeria) to 89% (Uganda); most countries reported COVID-19 vaccine acceptance at lower than 50% [3]. To our knowledge there have been no reports of COVID-19 vaccine impressions and beliefs to date from Chad, a country in north-central SSA [4].

From the time the first case of COVID-19 was confirmed in Chad in March 2020 [5] until March 2022, there have been more than 7,300 confirmed cases of COVID-19 and nearly 200 deaths reported [6]. It is likely the true number of cases is higher than the official count, due to issues with testing and reporting, especially early in the pandemic. Vaccination efforts in Chad began on 4 June 2021 after the donation of Sinopharm vaccines from China [7]. Through the COVAX program, Chad received a donation of Pfizer vaccines, bringing the country to 300,000 doses received by June 2021 [8]. The Chadian Health Minister has stated that "vaccination is open to all," and people over 65 years of age, medical staff, and candidates for the *Hajj* pilgrimage have been prioritized for immunization since the beginning of the vaccination campaign [7, 8]. From the beginning of its vaccination campaign to 1 April 2022, Chad has administered 419,469 COVID-19 vaccine doses—sufficient to fully vaccinate roughly one percent of the Chadian population [6].

Emerging data suggests that healthcare workers elsewhere in the African region (e.g., Ethiopia) endorse significantly high levels of COVID-19 vaccine hesitancy [9], even though healthcare workers may be presumed to be less vaccine hesitant than the general population. In a Zambian study, COVID-19 vaccine hesitancy was associated with one's beliefs about personal infectious risk [10]; this suggests that patients in hospitals, who tend to have more comorbidities and higher rates of immunocompromise than the general public, may be more accepting of the vaccine compared to the general population. For these reasons, we studied three main participant groups in Chad: healthcare workers, patients, and general community members. We also aimed to understand any common associations with vaccine hesitancy and acceptance since it is unclear if COVID-19 vaccine hesitancy relates to sociodemographic factors, education level, or household wealth in SSA. Studies to date in the SSA region have revealed conflicting results on important sociodemographic associations with COVID-19 vaccine hesitancy, some of which may be important to consider in order to roll out vaccinations successfully in Chad [11, 12]. Thus, understanding vaccine acceptance and hesitancy in Chad could inform public health campaigns and ultimately increase COVID-19 vaccine uptake if the information is acted upon. In this study, we aim to characterize vaccine acceptance and hesitancy in N'Djamena, Chad, including the associated COVID-19 and COVID-19 vaccine attitudes and beliefs

that inform these answers. To contextualize these findings, we report on the demographic, clinical, and socioeconomic associations with COVID-19 vaccine acceptance and hesitancy in Chadian patients, community members, and healthcare workers.

## Methods

### Ethics approval

The *Comité d'Ethique Hospitalier du Centre Hospitalier Universitaire de la Reference Nationale d'Djamena* reviewed and approved this study. The Mass General Brigham institutional review board also reviewed and approved this study (protocol #2021P001339). Informed consent was obtained verbally by Chadian study staff from each participant in their preferred language prior to survey commencement, and survey responses implied consent to participate. Separate written consent forms were not required.

### Setting

The United Nations Human Development Index ranks Chad 187th out of 189 countries [13], placing Chad in the low human development category. With a population of 16.4 million, the average life expectancy in Chad is 53 years for men and 56 years for women. Twenty-two percent of Chadians older than 15 years are literate, and the mean number of years of formal schooling is 2.5. Forty-two percent of the population lives below the national income poverty line in Chad [13]. The 2020 gross national income per capita (Atlas method) was 660 USD [14]. The official languages of Chad are French and Arabic, although there are many other local languages. N'Djamena is the capital city of Chad and is home to roughly 10% of the Chadian populace (2010 population of approximately 1.8 million) [15]. The University Hospital (National Reference Hospital [*translated*]), based in central N'Djamena, is one of teaching hospitals in Chad.

### Recruitment and enrollment

Participants were required to be ≥18 years old and residing in Chad. All participants in the patient group were recruited at the National Reference Hospital, N'Djamena. Participants in the community members group were recruited from public spaces, such as marketplaces, in the greater N'Djamena area. Participants in the healthcare workers group were recruited from both the National Reference Hospital and public spaces in greater N'Djamena.

### Study design and duration

This cross-sectional study was conducted from August to October 2021 using a one-time participant survey in a participant convenience sample, administered by Chadian study staff.

### Materials

The survey instrument queried participants on their sociodemographic characteristics; COVID-19 vaccine acceptance and vaccine hesitancy; past medical history, particularly as relevant to COVID-19; and attitudes and practices regarding COVID-19 and COVID-19 vaccination (see S1 Text). Vaccine hesitancy specifically was assessed using an adapted version of the WHO SAGE Vaccine Hesitancy Survey [16] (see S1 Text). Study staff, who were Chadian medical students (AB, ART, AG, and DF), administered the 25-question, 15-minute survey in-person to participants and recorded responses either on paper or electronically via *Qualtrics*. The survey was provided in French but administered by study staff in participants' local languages if they did not speak French.

## Participants and sample size

Participants were from three source populations: (1) patients at the National Reference Hospital, N'Djamena, (2) community members in public spaces in the greater N'Djamena area, and (3) healthcare providers recruited from both the National Reference Hospital and public spaces in greater N'Djamena. For this observational study, we set an arbitrary target of 500 participants based on available time and resources. Although we did not complete an *a priori* power calculation, based on an estimated 166 people per study group, there would be 80% power to detect at least a 13-point difference in two proportions with a level of significance of 0.05.

## Outcomes and data analysis

The primary outcomes were vaccine acceptance and vaccine hesitancy.

**Vaccine acceptance.** Participants were asked "If the COVID-19 vaccine were generally available, would you take the vaccine?" and responded "Yes" or "No."

**Vaccine hesitancy.** Participants responded to an adapted WHO SAGE Vaccine Hesitancy Survey (see S1 Text) [16]. We included the 10 binary and 10 Likert items from this survey. Items were modified to be relevant to the respondent and COVID-19. For example, the item "Do you believe that vaccines can protect children from serious diseases?" was adapted to "Do you believe that a vaccine can protect you from the COVID-19 disease?" A modified version of this survey for COVID-19 has already been validated and reported to demonstrate adequate psychometric properties [17]. Participants' responses to the 10 binary items were reported as proportions of participants in agreement or disagreement.

A sample statement from the 10 Likert items includes, "Having myself vaccinated for COVID-19 is important for the health of others in my community." Response options ranged from strongly disagree (score of 1) to strongly agree (score of 5) for these questions, seven of which were reverse scored. Scores from each of the 10 Likert items were summed to generate a composite Vaccine Hesitancy Scale (VHS) score ranging from 10 (least) to 50 (most) points. Consistent with previous literature [18], we also use a cutoff score of >30 to classify a subgroup of participants who were "highly hesitant." To evaluate the internal consistency of this scale, we calculated Cronbach's α.

Vaccine hesitancy was also analyzed qualitatively. After responding to the SAGE Vaccine Hesitancy Survey, we asked participants in an open-ended, free-response format "Please specify any other concerns you have about receiving a COVID-19 vaccine." Using an inductive thematic analysis approach, responses to these items were coded by the first author and codes were organized into themes. To establish inter-rater reliability, a second author reviewed the coding, and any discrepancies were discussed until resolved with agreement.

In addition to the primary outcomes, we queried participants on sociodemographic and clinical characteristics and attitudes and practices regarding COVID-19 and COVID-19 vaccination. These items (Tables 1 and 2) were considered as variables of interest potentially associated with vaccine acceptance and hesitancy in regression models.

Of these variables, one included an index—an adapted version of the Charlson comorbidity index (CCI) [19]. In our comorbidity index, the following self-reported comorbidities were included: cancer, diabetes mellitus, HIV, kidney disease, heart disease, liver disease, chronic lung disease, and cerebrovascular disease/stroke. These comorbidities were found to be significantly associated with mortality due to COVID-19 [20]. The index was dichotomized as the presence (versus not) of ≥1 comorbidity.

A chi-square test was used to analyze differences across groups in vaccine acceptance. Univariable regression models were fit with vaccine hesitancy scores as outcome variables and our

**Table 1.  Participant sociodemographic and clinical characteristics.**

| | Patients (n = 162) | Community members (n = 153) | Healthcare workers (n = 193) | Full cohort (n = 508) |
|---|---|---|---|---|
| Sex, n (%) | | | | |
| Female | 64 (40) | 46 (30) | 55 (28) | 165 (32) |
| Male | 98 (60) | 107 (70) | 138 (72) | 343 (68) |
| Age in years, mean (standard deviation; SD) | 36.2 (16.0) | 30.6 (11.3) | 29.2 (9.1) | 31.8 (12.6) |
| Highest level of education attained, n (%) | | | | |
| None | 26 (16) | 10 (7) | 1 (<1) | 37 (7) |
| Primary school | 13 (8) | 5 (3) | 4 (2) | 22 (4) |
| Secondary school | 12 (7) | 19 (12) | 3 (2) | 34 (7) |
| High school | 47 (29) | 28 (18) | 12 (6) | 87 (17) |
| University or beyond | 64 (40) | 91 (59) | 172 (89) | 327 (64) |
| Religious affiliation, n (%) | | | | |
| Christian | 109 (67) | 103 (67) | 109 (56) | 321 (63) |
| Muslim | 50 (31) | 47 (31) | 84 (44) | 181 (36) |
| Other | 3 (2) | 3 (2) | - | 6 (1) |
| Employment status, n (%) | | | | |
| Full-time | 40 (25) | 35 (23) | 25 (13) | 100 (20) |
| Part-time | 8 (5) | 19 (12) | 23 (12) | 50 (10) |
| Retired | 14 (9) | 2 (<1) | 1 (<1) | 17 (3) |
| Student | 50 (31) | 52 (34) | 121 (63) | 223 (44) |
| Unemployed | 50 (31) | 45 (29) | 21 (11) | 116 (23) |
| Average annual household income per capita,* mean in US dollars (SD) | 452 (599) | 694 (1215) | 657 (638) | 607 (843) |
| Household assets, n (%) | | | | |
| Electricity | 81 (50) | 111 (73) | 146 (76) | 338 (67) |
| Radio | 137 (85) | 134 (88) | 177 (92) | 448 (88) |
| Refrigerator | 38 (23) | 47 (31) | 81 (42) | 166 (33) |
| Television | 82 (51) | 103 (67) | 139 (72) | 324 (64) |
| Non-mobile telephone | 15 (9) | 8 (5) | 16 (8) | 39 (8) |
| Mobile telephone | 146 (90) | 146 (95) | 181 (94) | 473 (93) |
| Bicycle | 90 (56) | 67 (44) | 91 (47) | 248 (49) |
| Motorbike/scooter | 98 (60) | 102 (67) | 147 (76) | 347 (68) |
| Car | 29 (18) | 26 (17) | 74 (38) | 129 (25) |
| Truck | 7 (4) | 6 (4) | 16 (8) | 29 (6) |
| Past medical history, n (%) | | | | |
| Depression | 15 (9) | 12 (8) | 26 (13) | 53 (10) |
| Diabetes mellitus | 10 (6) | 7 (4) | 6 (3) | 23 (5) |
| Alcohol use | 28 (17) | 14 (9) | 21 (11) | 63 (12) |
| Tobacco use | 15 (9) | 5 (3) | 3 (2) | 23 (5) |
| Illicit drug use | - | 1 (<1) | 1 (<1) | 2 (<1) |
| Current pregnancy | 7 (4) | 5 (3) | 3 (2) | 15 (3) |
| Cerebrovascular disease/stroke | 3 (2) | 6 (4) | 4 (2) | 13 (3) |
| Hypertension | 26 (16) | 7 (5) | 7 (4) | 40 (8) |
| Headache or migraine | 104 (64) | 83 (54) | 88 (46) | 275 (54) |
| Seizures/epilepsy | 4 (2) | 2 (1) | 2 (1) | 8 (2) |
| Obesity | 2 (1) | 2 (1) | 4 (2) | 8 (2) |
| HIV | 2 (1) | 3 (2) | 1 (<1) | 6 (1) |
| Kidney disease | 8 (5) | 4 (3) | 6 (3) | 18 (4) |

(*Continued*)

**Table 1.** (*Continued*)

| | Patients (n = 162) | Community members (n = 153) | Healthcare workers (n = 193) | Full cohort (n = 508) |
|---|---|---|---|---|
| Heart disease | 1 (<1) | - | 1 (<1) | 2 (<1) |
| Liver disease | 1 (<1) | - | 3 (2) | 4 (<1) |
| Cancer | - | - | - | - |
| Chronic lung disease | 2 (1) | 2 (1) | 3 (2) | 7 (1) |
| Adapted Charlson comorbidity index (CCI), n with ≥1 comorbidity (%) | 25 (15) | 20 (13) | 19 (10) | 64 (13) |

*As of November 2021, 1 US dollar = 577.11 Central African CFA Francs

demographic, social, and clinical variables of interest as explanatory variables. Multivariable models were constructed to assess associations with vaccine hesitancy ($p < .20$ in the univariable regressions). Sensitivity analyses were conducted to determine if the findings persisted after excluding participants who had already been vaccinated prior to participation. To explore the relationship between vaccine acceptance and hesitancy, a student's *t*-test was conducted on VHS scores between groups who responded "Yes" versus "No" on the acceptance measure, which additionally provides an analysis of concurrent validity of the VHS. A two-tailed *p*-value $< .05$ was considered statistically significant.

## Results

### Participant characteristics

We enrolled 516 participants. After excluding participants with incomplete data for the VHS (n = 8), data from 508 participants—comprising 162 patients, 153 community members, and 193 healthcare workers—were analyzed.

Among patients, the most common specialties where care was sought were: infectious disease (n = 57; most commonly for malaria, n = 37), neurology (n = 54; most commonly for epilepsy, n = 20), emergency medicine (n = 15, most commonly for gunshot wounds, n = 3, and

**Table 2. Participant attitudes and practices regarding COVID-19.**

| | Patients (n = 162) | Community members (n = 153) | Healthcare workers (n = 193) | Full cohort (n = 508) |
|---|---|---|---|---|
| Vaccination received, n (%) | | | | |
| AstraZeneca | 1 (<1) | 1 (<1) | 1 (<1) | 3 (<1) |
| Pfizer-BioNTech | 2 (1) | 3 (2) | 11 (6) | 16 (3) |
| Sinopharm | 28 (17) | 8 (5) | 17 (9) | 53 (10) |
| Unknown | 2 (1) | 7 (4) | 5 (3) | 14 (3) |
| Ever tested for COVID-19, n (%) | 34 (21) | 39 (24) | 37 (19) | 110 (22) |
| Ever positive for COVID-19 | 13 (8) | 9 (6) | 10 (5) | 32 (6) |
| Had symptoms of COVID-19 in past 12 months, n (%) | | | | |
| Fever | 104 (64) | 71 (46) | 110 (57) | 285 (56) |
| Tiredness | 74 (46) | 69 (43) | 108 (56) | 251 (49) |
| Dry cough | 52 (32) | 29 (18) | 47 (24) | 128 (25) |
| Aches and pains | 40 (25) | 35 (22) | 55 (28) | 130 (26) |
| Nasal congestion | 22 (14) | 24 (15) | 56 (29) | 102 (20) |
| Loss of smell | 16 (10) | 9 (6) | 26 (13) | 51 (10) |

(*Continued*)

**Table 2.** (Continued)

| | Patients (n = 162) | Community members (n = 153) | Healthcare workers (n = 193) | Full cohort (n = 508) |
|---|---|---|---|---|
| Loss of taste | 13 (8) | 11 (7) | 26 (13) | 50 (10) |
| Headache | 38 (23) | 29 (19) | 48 (25) | 115 (23) |
| Runny nose | 46 (28) | 39 (25) | 60 (31) | 145 (29) |
| Shortness of breath | 8 (5) | 5 (3) | 12 (6) | 25 (5) |
| Sore throat | 28 (17) | 12 (8) | 28 (15) | 68 (13) |
| Pneumonia | 4 (2) | 4 (3) | 10 (5) | 18 (4) |
| Diarrhea | 44 (27) | 33 (22) | 38 (20) | 115 (23) |
| Vaccine acceptance, n (%) | 108 (67) | 68 (44) | 90 (47) | 266 (52) |
| SAGE Vaccine Hesitancy Scale score | | | | |
| Mean (SD) | 27.0 (7.7) | 28.9 (10.6) | 29.4 (10.0) | 28.5 (9.6) |
| Highly hesitant (score >30), n (%) | 34 (21) | 63 (41) | 87 (45) | 184 (36) |
| Knows someone who has..., n (%) | | | | |
| been hospitalized for COVID-19 | 72 (44) | 50 (33) | 98 (51) | 220 (43) |
| died due to COVID-19 | 63 (39) | 41 (27) | 66 (34) | 170 (33) |
| Believes community leaders support COVID-19 vaccination, n (%) | | | | |
| Religious leaders | 109 (67) | 107 (70) | 109 (56) | 325 (64) |
| Political leaders | 136 (84) | 135 (88) | 160 (83) | 431 (85) |
| Teachers | 144 (89) | 124 (81) | 140 (73) | 408 (80) |
| Healthcare workers | 150 (93) | 140 (92) | 151 (78) | 441 (87) |
| Concerns about taking the COVID-19 vaccine, n yes (%) | | | | |
| None | 80 (50) | 56 (37) | 57 (30) | 193 (38) |
| Vaccine efficacy | 61 (38) | 55 (36) | 76 (39) | 192 (38) |
| Vaccine safety | 55 (34) | 49 (32) | 69 (36) | 173 (34) |
| Religious concerns | 8 (5) | 9 (6) | 14 (7) | 31 (6) |
| Fear of needles | 24 (15) | 14 (9) | 20 (10) | 58 (11) |
| Don't think COVID-19 presents a risk to me | 17 (10) | 19 (12) | 31 (16) | 67 (13) |
| Bad experience with previous vaccination | 6 (4) | 12 (8) | 9 (5) | 27 (5) |
| Concerns about the pharmaceutical industry | 37 (23) | 38 (25) | 62 (32) | 137 (27) |
| Don't trust the government | 30 (41) | 24 (16) | 51 (26) | 105 (21) |
| Concerns from TV/radio/news | 9 (6) | 11 (7) | 26 (13) | 46 (9) |
| Concerns from social media | 9 (6) | 15 (10) | 25 (13) | 49 (10) |
| Transportation | 9 (6) | 13 (8) | 18 (9) | 40 (8) |
| Cost | 7 (4) | 6 (4) | 9 (5) | 22 (4) |
| Side effects | 77 (48) | 65 (42) | 102 (53) | 244 (48) |
| Prefer traditional healing | 5 (3) | 6 (4) | 16 (8) | 27 (5) |
| Where COVID-19 information is sought, n (%) | | | | |
| Television | 113 (70) | 122 (80) | 172 (89) | 407 (80) |
| Radio | 141 (87) | 138 (90) | 169 (88) | 448 (88) |
| Social media | 98 (60) | 105 (69) | 154 (80) | 357 (70) |
| Local organizations | 58 (36) | 56 (37) | 84 (44) | 198 (39) |
| International organizations (such as WHO) | 63 (39) | 58 (38) | 117 (61) | 238 (47) |
| Healthcare providers | 47 (30) | 46 (30) | 82 (42) | 175 (34) |
| Traditional healers | 10 (6) | 13 (8) | 9 (5) | 32 (6) |
| Friends | 116 (72) | 105 (69) | 139 (72) | 360 (71) |
| Family | 109 (67) | 95 (62) | 113 (59) | 317 (62) |
| Co-workers | 42 (26) | 59 (39) | 119 (62) | 220 (43) |

diabetic emergencies, n = 3), primary care (n = 14, most commonly for diabetes, n = 6), cardiology (n = 12, most commonly for hypertension, n = 10), and rheumatology (n = 10).

Community members were most commonly students (n = 52), merchants and businesspeople (n = 41), housewives and caretakers (n = 20), teachers (n = 14), agricultural workers (n = 6), engineers and scientists (n = 6), taxi drivers and chauffeurs (n = 5), artists and musicians (n = 5), and athletes (n = 5). Other occupations represented include law enforcement workers, mechanics, tailors, airline workers, traditional healers, clergy people, and construction workers.

Healthcare workers were medical students or students in other healthcare fields (n = 121), allied workers (n = 23; such as laboratory staff), physicians (n = 16), medical instructors (n = 12), nurses (n = 9), healthcare administrators (n = 7), pharmacy staff (n = 3), and midwives (n = 2).

The average age of participants was 32 years (range 18–90 years), and all participants identified as African Black. Roughly two-thirds of participants were male. Most of the sample was highly educated: 81% had attained at least a high school level of education—nearly half of this group was comprised of healthcare workers. Most participants were either students or full-time workers (64%, n = 323). Thirteen percent of participants (n = 64) had at least one comorbidity according to the CCI, the most common of which were diabetes, kidney disease, and cerebrovascular disease/stroke.

The average household income per capita was the equivalent of 607 USD, which is comparable to the 2020 Chadian gross national income per capita of 660 USD (as of November 2021, 1 US dollar = 577.11 Central African CFA Francs). Regarding household assets, most participants lived in households with electricity, a radio, a television, a mobile phone, and a motorbike/scooter. Less than a third of participants had a car or truck, and less than a third had a refrigerator in their household.

Additional participant characteristics are found in Table 1.

## Attitudes and practices regarding COVID-19

Of all participants, 22% (n = 110) had ever been tested for COVID-19; of them, 29% (n = 32) had ever tested positive for COVID-19—representing 6% of the entire sample. Participants were queried on any symptoms of COVID-19 they had in the past 12 months. The symptoms most specific to COVID-19, anosmia (loss of smell) and ageusia (loss of taste), were each endorsed by 10% of participants in the past 12 months (n = 51 and n = 50, respectively; Table 2). Approximately half of participants responded that they knew anyone who had ever been hospitalized for COVID-19, and approximately one-third responded that they knew someone who had died due to COVID-19.

Participants were asked whether they believed their community leaders—i.e., religious leaders, political leaders, teachers, and healthcare workers—supported COVID-19 vaccination. Participants' perceptions of leader support of COVID-19 vaccination were high, ranging from 64% to 87%: participants were least likely to believe religious leaders supported vaccination (64%) and most likely to believe healthcare workers supported vaccination (87%).

As part of the WHO SAGE Vaccine Hesitancy Survey, participants were presented a list of possible concerns they may have regarding vaccination and asked to select any that were personally applicable. One-hundred and ninety-three participants (38%) responded that they have no concerns about taking the vaccine. Concerns endorsed by at least 10% of the sample included: vaccine side effects (n = 244, 48%), vaccine efficacy (n = 192, 38%), vaccine safety (n = 173, 34%), concerns about the pharmaceutical industry (n = 137, 27%), lack of trust in the government (n = 105, 21%), belief that COVID-19 does not present a personal risk (n = 67, 13%), and fear of needles (n = 58, 11%).

When asked where they sought information regarding COVID-19, radio (88%) and television (80%) predominated. More than half of participants sought information through friends and family (71% and 62% respectively) and social media (70%). Participants were less likely to seek information from their healthcare providers (34%), local organizations (34%), or traditional healers (6%).

## Vaccine acceptance and vaccine hesitancy

**Vaccine acceptance.** When asked explicitly if they would take the vaccine if available, 52% of participants responded affirmatively (n = 266). This varied slightly across groups: 67% in patients, 44% in community members, and 47% in healthcare workers. Vaccine acceptance was statistically significantly higher among patients than among community members and healthcare workers, $p < .001$. Of note, 86 participants had already been vaccinated at the time of survey administration.

**Vaccine hesitancy.** The VHS demonstrated high internal consistency ($\alpha = .81$). The mean score on the VHS for all participant groups was 28.5 points (SD = 9.6 points); this varied slightly from 27.0 points in patients to 29.4 points in healthcare workers. Using a cutoff of $\geq 30$ points, 21% of patients were considered highly hesitant, alongside 41% of community members and 45% of healthcare workers.

Within the 10 binary items of the Vaccine Hesitancy Survey (Fig 1), the three items for which participants endorsed the most vaccine hesitancy were: having heard negative information about the COVID-19 vaccination (72% of participants), believing it is difficult for some ethnic or religious groups in their community to get vaccinations (59%), and not thinking that "most people like you will take the COVID-19 vaccine" (50%). The three items in which participants endorsed the least vaccine hesitancy were: not having other pressures in life preventing them from getting the COVID-19 vaccine (70%), never having refused a different vaccine in the past (69%), and believing a vaccine can protect them from the COVID-19 disease (64%).

Within the 10 Likert items (Fig 1), the three items for which participants endorsed the most vaccine hesitancy (i.e., the items with the most "agree" and "strongly agree" responses) were: being concerned about serious adverse effects of the vaccine (61%), believing there is no need to be vaccinated because COVID-19 does not present a personal risk (42%), and believing that COVID-19 vaccines carry more risks than older vaccines (41%). The three items with the least hesitancy were: believing personal vaccination for COVID-19 is important for the health of others in the community (75%), believing COVID-19 vaccines are effective (74%), and generally doing what their doctor or health care provider recommends about vaccines (74%).

The mean VHS score of participants responding "No" to the vaccine acceptance question was significantly higher (36.1 points) than the mean score for those responding "Yes" (22.6 points), $p < .001$, which supports the scale's concurrent validity.

## Associations of vaccine hesitancy

Table 3 depicts variables included in univariable regression models with associated statistics. In the univariable models, healthcare workers had statistically significantly higher vaccine hesitancy ($p = .017$) than other participant types. Older age was associated with less vaccine hesitancy ($p < .001$). There were no differences in vaccine hesitancy by sex, educational status, or religion. Participants who were employed full-time reported lower levels of vaccine hesitancy than participants of other employment statuses ($p = .015$). Other variables statistically significantly associated with lower levels of vaccine hesitancy included knowing someone who had died from COVID-19 ($p < .001$), believing one's local healthcare workers support vaccination ($p < .001$), having medical comorbidities ($p = .001$), having a higher socioeconomic status

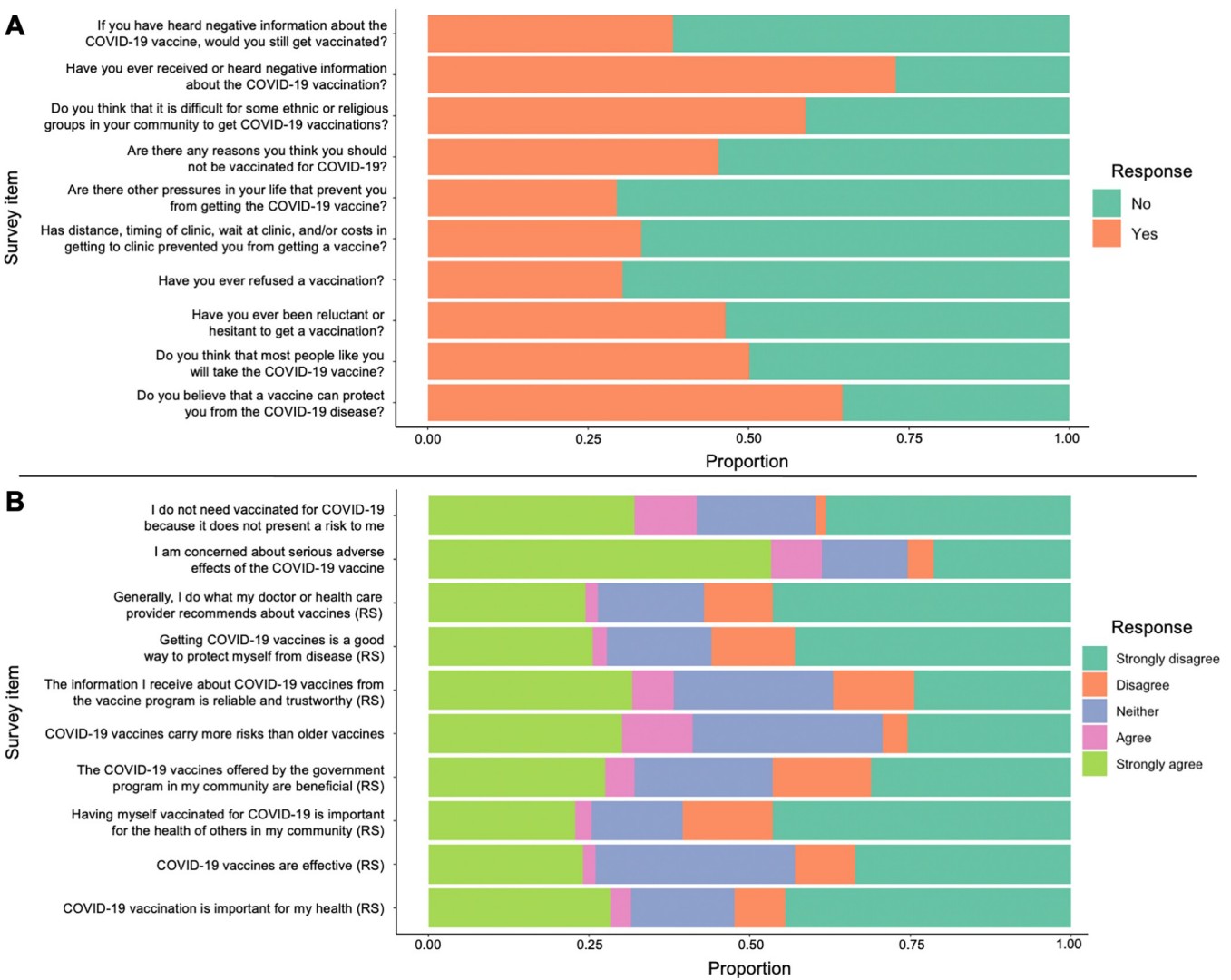

**Fig 1. Distribution of participant responses to the 10 binary (Panel A) and 10 Likert items (Panel B) of the modified WHO SAGE Vaccine Hesitancy Survey.** RS suggests the item was reverse scored for the Vaccine Hesitancy Scale.

(operationalized here as living in a household with electricity; p = .016), and trusting the government regarding COVID-19 (p < .001).

In the multivariable model (Table 4), knowing someone who died from COVID-19, believing local healthcare workers support vaccination, trusting the government regarding COVID-19, having a higher socioeconomic status (having electricity), and having medical comorbidities were each associated with less vaccine hesitancy (all p < .05).

A sensitivity analysis, removing participants in the sample who had already been vaccinated for COVID-19, yielded nearly identical results, except for the household assets (having electricity) and government trust variables, which were no longer statistically significantly associated with vaccine hesitancy (p = .148 and p = .060, respectively). We also created both univariable and multivariable models with vaccine acceptance as the response variable, which yielded results nearly identical to the models for vaccine hesitancy.

**Table 3. Univariable linear regressions evaluating variables associated with vaccine hesitancy in Chad.**

| Variable | Mean VHS score | Denominator degrees of freedom | β [95% confidence intervals] | p |
|---|---|---|---|---|
| Participant type | | 505 | | |
| Patient (reference category) | 27.0 | | | |
| Community member | 28.9 | | 1.88 [-0.23, 3.99] | .082 |
| Healthcare worker | 29.4 | | 2.43 [0.44, 4.43] | .017 |
| Sex | | 506 | | |
| Female (reference category) | 29.2 | | | |
| Male | 28.2 | | -1.00 [-2.78, 0.78] | .272 |
| Age (continuous) | - | 503 | -0.11 [-0.18, -0.05] | < .001 |
| Highest level of education attained | | 506 | | |
| Less than high school (reference category) | 28.2 | | | |
| High school or beyond | 28.6 | | 0.37 [-1.78, 2.52] | .737 |
| Employment status | | 501 | | |
| Student (reference category) | 29.2 | | | |
| Full-time employment | 26.5 | | -2.79 [-5.04, -0.54] | .015 |
| Part-time employment | 28.0 | | -1.26 [-4.18, 1.67] | .399 |
| Retired | 24.7 | | -4.53 [-9.24, 0.18] | .059 |
| Unemployed | 29.6 | | 0.41 [-1.73, 2.55] | .708 |
| Religion | | 504 | | |
| Animist (reference category) | 27.2 | | | |
| Christian | 29.0 | | 1.82 [-6.66, 10.30] | .673 |
| Muslim | 27.6 | | 0.38 [-8.15, 8.91] | .930 |
| Knows someone who has died due to COVID-19 | | 468 | | |
| No (reference category) | 29.8 | | | |
| Yes | 25.2 | | -4.65 [-6.41, -2.90] | < .001 |
| Seeks COVID-19 information from television | | 506 | | |
| No (reference category) | 28.5 | | | |
| Yes | 28.5 | | 0.06 [-2.04, 2.15] | .957 |
| Seeks COVID-19 information from social media | | 506 | | |
| No (reference category) | 28.9 | | | |
| Yes | 28.3 | | -0.53 [-2.36, 1.30] | .569 |
| Believes local healthcare workers support vaccination | | 504 | | |
| No (reference category) | 33.4 | | | |
| Yes | 27.8 | | -5.60 [-8.06, -3.15] | < .001 |
| Comorbidities index | | 506 | | |
| No comorbidities (reference category) | 29.0 | | | |
| ≥1 comorbidity | 24.9 | | -4.17 [-6.66, -1.68] | .001 |
| Household assets (electricity) | | 500 | | |
| No (reference category) | 29.9 | | | |
| Yes | 27.7 | | -2.19 [-3.97, -0.41] | .016 |
| Trusts the government regarding COVID-19 | | 506 | | |
| No (reference category) | 31.3 | | | |
| Yes | 27.8 | | -3.57 [-5.61, 1.53] | < .001 |

## Qualitative vaccination concerns

One hundred and sixteen participants provided responses to the open-ended measure regarding concerns about the COVID-19 vaccine. Four main themes were identified in participants' responses: 1) education, 2) trust, 3) clinical concerns, and 4) misinformation and false beliefs.

**Table 4. Multivariable linear regression statistics evaluating variables associated with vaccine hesitancy in Chad.**

| Variable | β | 95% confidence interval | p |
|---|---|---|---|
| Participant type | | | |
| Patient (reference category) | | | |
| Community member | 1.63 | [-0.59, 3.84] | .150 |
| Healthcare worker | 1.17 | [-1.04, 3.39] | .299 |
| Age | -0.03 | [-0.12, 0.06] | .545 |
| Employment status | | | |
| Student (reference category) | | | |
| Full-time employment | -0.51 | [-3.19, 2.17] | .709 |
| Part-time employment | -0.21 | [-3.23, 2.82] | .893 |
| Retired | -0.13 | [-6.23, 5.97] | .967 |
| Unemployed | 0.91 | [-1.46, 3.28] | .452 |
| Knows someone who has died due to COVID-19 | -3.79 | [-5.56, -2.01] | < .001* |
| Believes local healthcare workers support vaccination | -5.28 | [-7.85, -2.70] | < .001* |
| ≥1 comorbidity | -2.91 | [-5.40, -0.41] | .023* |
| Household assets (electricity) | -2.17 | [-4.06, -0.28] | .025* |
| Does not trust the government regarding COVID-19 | 2.76 | [0.66, 4.85] | .010* |

*p < .05

## Education

Ten participants (9%) stated a lack of personal knowledge about the vaccine as a vaccination concern. For example, a 24-year-old female patient was unsure how to decide between different vaccine manufacturers: "I don't know which of the vaccines is reliable." Another patient, a 21-year-old man, reported: "I don't know the constitution of the vaccine." This theme was also prevalent among a few reporting healthcare workers, such as one 63-year-old man who was hesitant about getting the vaccine because he didn't know "how long [the vaccine] protects us."

## Trust

Twenty participants (17% of respondents to the open-ended measure) were concerned about taking the COVID-19 vaccine because they did not trust a variety of sources, primarily capitalism (i.e., believing that business and capitalism are controlling the pandemic; n = 9), pharmaceutical companies (n = 7), and White people and high-income countries (n = 4). For example, a 25-year-old male healthcare worker stated, "Vaccination against COVID-19 is purely political in my opinion. Developed countries want to make money." A 25-year-old male community member said, "The corona vaccine is that which whites have created to eliminate us." Finally, a 28-year-old male healthcare worker mentioned that the "[v]accine [was] manufactured by pharmaceutical firms to enrich themselves."

## Clinical concerns

Forty-eight participants (41%) discussed clinical concerns they had regarding COVID-19 vaccine, including side effects (n = 27), vaccine efficacy (n = 11), and clinical research on the vaccine (n = 10). In addition to the many participants who were broadly concerned about vaccine side effects, several cited specific side effect concerns, especially cerebral thrombosis, cerebral edema, and coagulopathy. Notably, participants who discussed side effect concerns were a mix of patients, community members, and healthcare workers. Likewise, of the 11 participants

who cited vaccine efficacy/effectiveness as a concern, there was a mix of patients, community members, and healthcare providers. Regarding clinical research concerns, participants had a wide array of thoughts. Multiple participants believed the vaccine was developed "too fast," and several claimed it takes 10 years before a vaccine can be effective. For example, one 45-year-old male patient said, "It is not yet well elucidated to talk about its effectiveness, it takes at least 10 years first." A 47-year-old male patient discussing clinical research on the vaccine claimed that "Manufacturing steps were burned." Finally, a 27-year-old female healthcare worker believed that "[t]his vaccine is not certified [as] there have been no clinical trials on it."

## Misinformation and false beliefs

Almost half of respondents (n = 57, 49%) endorsed false beliefs about COVID-19 and/or the COVID-19 vaccine. Thirty-two believed that COVID-19 was a hoax, that it was witchcraft, or that it simply did not exist. Ten participants acknowledged the existence of COVID-19 but believed that vaccination was not necessary. For example, a 32-year-old female community member believed, "We are already living with COVID-19, so [there is] no need to vaccinate because we are immune naturally." Another participant (34-year-old male community member) claimed that vaccination was not necessary because "Chadians have an effective antibiotic which is rock salt, commonly called 'le chou.'" One participant (22-year-old male community member) mentioned, "When we don't feel sick, we don't need a vaccine," and yet another (22-year-old female patient) mentioned "I didn't have COVID-19, so why take the vaccine?"

Fifteen participants directly endorsed misinformation about the COVID-19 vaccine and its outcomes. Several endorsed the claim that the vaccine contains strains of different viruses (commonly HIV and Ebola), such as one 35-year-old female patient who said it "[a]ppears that this vaccine contains the HIV virus and that anyone vaccinated will have AIDS in 2 years." A 32-year-old female healthcare worker stated, "Deceased after 2 years of vaccination, inoculation of the HIV germ in the body of the vaccinated person." One participant reported hearing a talk regarding this issue: "The COVID-19 vaccine is associated with the AIDS virus, I listened to a talk about this" (47-year-old female healthcare worker). Other commonly endorsed false beliefs included decreased fertility and life expectancy.

## Discussion

We establish an evidence base for COVID-19 vaccine acceptance and hesitancy in N'Djamena, Chad using structured surveys of patients, community members, and healthcare workers. In this convenience sample, vaccine acceptance approximated 52%, and reasons for vaccine hesitancy were broad—spanning from concerns on vaccine side effects and efficacy to believing misinformation about the COVID-19 vaccine to not trusting the government and institutions in charge of vaccine research, development, and administration. We identified subgroups of participants who were more likely to endorse vaccine hesitancy, such as those who did not (i) personally know anyone who died from COVID-19, (ii) believe their local healthcare workers supported vaccination, (iii) trust the government regarding COVID-19, (iv) have medical comorbidities, or (v) have electricity in their household (i.e., those who had a lower socioeconomic status).

Only approximately half of Chadian participants reported that they would accept the COVID-19 vaccine. Compared to 2020 data from the Africa Centres for Disease Control and Prevention (Africa CDC), this is substantially lower than for 15 other countries reporting in the SSA region, where the average vaccine acceptance rate was nearly 80% and the lowest reported rate was 59% in the Democratic Republic of Congo [21]. However, compared to a recent preprint of a scoping review, which included data from various countries in SSA from 1

January 2020 to 5 July 2021, our data are in line with the majority of countries that had vaccine acceptance rates lower than 50% [3].

In accordance with data from other SSA countries, suggesting nearly two-thirds of people identify television and radio as some of their most trusted sources of information [21], television and radio were the most highly utilized sources of COVID-19 information in our sample. This highlights television and radio as outlets with widespread reach that may be targeted for public health messaging. However, 70% of respondents, who were on average young, report use of social media as a source of vaccine information, highlighting this platform also as a potential outlet for vaccination outreach.

More than one-third of participants were classified as highly hesitant. Interestingly, this statistic was 45% among healthcare workers—a group that might be expected to have the lowest rates of vaccine hesitancy given their field of work and training. This rate however aligns with the prevalence of vaccine hesitancy among healthcare workers in Zimbabwe, Ghana, South Africa, Kenya, Sudan, and Ethiopia [22] and highlights the importance of including healthcare workers in public health outreach throughout COVID-19 vaccination campaigns.

Of all measured variables possibly associated with vaccine hesitancy in Chad, the psychosocial variables predominated as statistically significantly associated, including knowing someone who died from COVID-19, believing one's local healthcare workers support vaccination, and trusting the government regarding COVID-19. Because participants' perceptions of local healthcare workers' beliefs were significantly associated with vaccine hesitancy, utilizing local community healthcare workers may be a viable strategy for vaccination campaigns to increase uptake in Chadian communities.

The previous study of 15 countries in SSA conducted by the Africa CDC found a significant association between more vaccine hesitancy and social media use [21]. While more than two-thirds of participants in our study reported seeking COVID-19 information through social media use, we did not find a significant association between higher levels of vaccine hesitancy and use of social media, where vaccine mis- and dis-information are likely to prevail. We also did not find significant associations with other variables previously reported to be associated with COVID-19 vaccine hesitancy in SSA, such as participant sex or religious beliefs [21, 22]. An important characteristic of our study population is the high number of participants who had at least a high school education (nearly 80%) which is much higher than the proportion in Chad as a nation. This made variations among people with less than a high school education too few to fully analyze in this study. Also, of note, our survey queried participants on their religious identity only (e.g., none, Muslim, Christian, etc.), not their religiosity. Given that the belief that prayer is more important than vaccination in preventing COVID-19 infection is a significant factor in overwhelmingly large proportions of people in Niger (89%), Liberia (86%), and Senegal (71%), religiosity remains an important, unstudied aspect of vaccine acceptance and hesitancy [23].

One of the themes that emerged in the qualitative data regarding vaccine hesitancy was that of trust. Participants mentioned they were concerned about taking the COVID-19 vaccine because of lack of trust in pharmaceutical companies, Western societies, and White people. This is also in line with the review conducted by the Africa CDC, in which 43% of respondents believed that "People in Africa are being used as guinea pigs in vaccine trials" and almost half of respondents believed that COVID-19 was "a planned event by foreign actors" [21]. Furthermore, the only other literature specifically focusing on vaccine hesitancy in Chad, which reports on general vaccine hesitancy (including the polio vaccine) among mobile pastoralist communities in Chad in 2016 [24], identified trust as the most frequently reported barrier to vaccination—suggesting that trust issues regarding vaccination in Chad are not limited to the COVID-19 vaccine and merit further intervention.

Another theme identified in the qualitative data included clinical concerns, which overlapped with the quantitative data reported here and largely comprised concerns about vaccine side effects and vaccine efficacy. This theme along with the education theme, in which participants largely reported a lack of personal knowledge about the vaccine as a reason for vaccine hesitancy, and the misinformation/false beliefs theme, in which participants endorsed many false beliefs about the COVID-19 vaccine comprise major aspects of vaccine hesitancy in Chad that are addressable. Providing data-driven evidence and education about COVID-19 and the vaccine to the populace could have a significant impact on vaccine uptake.

Our research had limitations. Because this was a convenience sample in an urban area, we are unable to make population-based assumptions about vaccine hesitancy in Chad nationally. Furthermore, our recruitment technique did not allow us to calculate a response rate for all individuals who were approached to participate in the research and complete the survey; thus, there may have been a selection bias in our sample. Indeed, members of the study team in Chad mentioned that some individuals approached would not participate because they "didn't like to hear anything regarding COVID-19," others because the study included American collaborators, and others because they were afraid the government might gain access to their responses. Although the average income in our participants was comparable to the average income in Chad, another limitation of the current study was the skewed education level of participants: more than three-fourths of our participants had attained at least a high school level of education. Despite this high educational attainment level overall, our findings demonstrate that vaccine hesitancy is common in Chad.

Our study also had several strengths. We provide the first account of vaccine acceptance and hesitancy in Chad, where minimal COVID-19 vaccination research has been conducted. We also achieved a large sample size which enabled well-powered statistical analyses for associations with our outcomes of interest. The use of both quantitative and qualitative data allowed us to report both broad findings and more nuanced discussions of individuals' concerns regarding COVID-19 vaccination. Future efforts could utilize the findings presented here to inform public health education and vaccination campaign interventions in Chad to achieve widespread uptake in the country. For example, for other vaccines in Chad, such as the rabies and polio vaccines, call centers and hotlines have been used to field individuals' questions and concerns regarding vaccination [25, 26]. Similarly, call centers with Chadian staffers who are educated on the specific vaccination concerns presented in the current study could be considered.

Since COVID-19 and COVID-19 vaccine hesitancy are dynamic [27], future research should again quantify vaccine hesitancy in Chad and examine the potential impact of disease spread, public health messages, and other efforts. An analysis of early COVID-19 vaccination uptake in 15 countries in West Africa [28] suggested that to reach targeted coverage of 60% of the population within 12 months, vaccination uptake would need to increase by 7 times the current trajectory. Although many barriers exist to this pace of vaccination uptake in SSA (such as vaccine access [29], community engagement [30], travel to vaccination sites [29], etc.), addressing the issues of vaccine hesitancy reported here presents one avenue through which vaccination uptake may be bolstered. As COVID variants and other disease outbreaks remain ongoing global threats, formative work on vaccine beliefs and addressing specific vaccine concerns remain as important as vaccine access itself.

## Supporting information

**S1 Checklist. Research reporting checklist.**
(DOCX)

**S1 Text. Survey instrument used for this study, including the adapted version of the WHO SAGE VHS.**
(DOCX)

**S1 Data. De-identified dataset.**
(XLSX)

**S1 File. Required questionnaire on research inclusivity.**
(DOCX)

## Author Contributions

**Conceptualization:** Dylan R. Rice, Farrah J. Mateen.

**Data curation:** Anatole Balamo, Allah-Rabaye Thierry, Aremadji Gueral, Djerakoula Fidele, Foksouna Sakadi.

**Formal analysis:** Dylan R. Rice.

**Funding acquisition:** Farrah J. Mateen.

**Investigation:** Dylan R. Rice, Farrah J. Mateen.

**Methodology:** Dylan R. Rice, Farrah J. Mateen.

**Project administration:** Dylan R. Rice, Anatole Balamo, Allah-Rabaye Thierry, Aremadji Gueral, Djerakoula Fidele, Farrah J. Mateen, Foksouna Sakadi.

**Resources:** Farrah J. Mateen.

**Supervision:** Farrah J. Mateen.

**Validation:** Farrah J. Mateen.

**Visualization:** Dylan R. Rice.

**Writing – original draft:** Dylan R. Rice.

**Writing – review & editing:** Dylan R. Rice, Farrah J. Mateen.

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
