## [Decision Letter · Decision Letter 0]

17 Mar 2022

PGPH-D-22-00016

COVID-19 vaccine acceptance and hesitancy in Chad: A cross-sectional study of patients, community members, and healthcare workers

Dear Dr. Mateen,

Thank you for submitting your manuscript to PLOS Global Public Health. After careful consideration, we feel that it has merit but does not fully meet PLOS Global Public Health’s publication criteria as it currently stands. Therefore, we invite you to submit a revised version of the manuscript that addresses the points raised during the review process.

We look forward to receiving your revised manuscript.

Kind regards,

Joel Msafiri Francis, MD, MS, PhD

Academic Editor

Journal Requirements:

2. Please amend your Financial Disclosure statement. If you did not receive any funding for this study, please simply state: “The authors received no specific funding for this work.”

3. Please update your Competing Interests statement. If you have no competing interests to declare, please state: “The authors have declared that no competing interests exist.”

4. In the online submission form, you indicated that “De-identified data will be made available upon request by qualified investigators.”. All PLOS journals now require all data underlying the findings described in their manuscript to be freely available to other researchers, either 1. In a public repository, 2. Within the manuscript itself, or 3. Uploaded as supplementary information.

5. We have noticed that you have uploaded supporting information but you have not included a list of legends.  Please add a full list of legends for all supporting information files (including figures, table and data files) after the references list.

Additional Editor Comments (if provided):

It would be more helpful to use the STROBE guidelines (see below) in structuring the paper.

https://www.equator-network.org/reporting-guidelines/strobe/

Reviewers' comments:

Reviewer's Responses to Questions

**Comments to the Author**

1. Does this manuscript meet PLOS Global Public Health’s publication criteria? Is the manuscript technically sound, and do the data support the conclusions? The manuscript must describe methodologically and ethically rigorous research with conclusions that are appropriately drawn based on the data presented.

Reviewer #1: Partly

Reviewer #2: Partly

Reviewer #3: Yes

2. Has the statistical analysis been performed appropriately and rigorously?

Reviewer #1: Yes

Reviewer #2: Yes

Reviewer #3: Yes

3. Have the authors made all data underlying the findings in their manuscript fully available (please refer to the Data Availability Statement at the start of the manuscript PDF file)?

Reviewer #1: Yes

Reviewer #2: Yes

Reviewer #3: No

4. Is the manuscript presented in an intelligible fashion and written in standard English?

Reviewer #1: Yes

Reviewer #2: Yes

Reviewer #3: Yes

5. Review Comments to the Author

Reviewer #1: In the submitted manuscript titled: COVID-19 vaccine acceptance and hesitancy in Chad: A cross-sectional study of patients, community members, and healthcare workers”, the authors described the situation of COVID-19 vaccine uptake among different groups in Chad.

Below are points that need to be clarified

1. Abstract: Please include the main aim of this study in the introduction section of the abstract

2. Introduction:

a. First paragraph: You may wish to give a global picture of COVID-19 vaccine uptake and hesitancy, then narrow down

b. “Since the beginning of Chad’s vaccination campaign, people over 65 years of age, medical staff, and candidates for the Hajj pilgrimage have been prioritized for immunization, although the Chadian Health Minister has stated that “vaccination is open to all.” This statement is contradictory, kindly rephrase to fit your manuscript

c. “From the beginning of its vaccination campaign to 12 December 2021, Chad has administered 285,922 COVID-19 vaccine doses—sufficient to fully vaccinate less than one percent of the Chadian population” This sentence may be moved to the above paragraph as it discusses similar concepts as in para 1, the remaining sentences can be merged with para 3

d. “One factor at the individual level preventing COVID-19 vaccine uptake is vaccine hesitance,” This statement sounds controversial, as vaccine hesitancy is a broad term by itself. Please clarify this statement in relation to the reference cited

e. Please reduce the number of preprints cited as there are a lot of published papers in covid-19 vaccine acceptance and hesitancy

f. “a country with a very young population and with lower average per capita wealth than many other Sub-Saharan African countries” Maybe not necessary to be added in this specific section

g. “In this study, we aim to characterize vaccine acceptance and hesitancy in Chad and report on demographic, clinical, and socioeconomic associations “ A brief review of literature on these factors may be added

h. Please include your aim of conducting this study towards the end of the introduction section

i. In the introduction I have failed to capture the reason of the choice of the study population in this research. Please include a paragraph explaining why you chose health care workers, patients and community members. How are these populations linked? Any rationale of such grouping? Please back up using literature in each specific group

3. Methods

a. Kindly check the journal style if ethics approval comes first in the section

b. Please include ethical approval numbers for each institution

c. “Participants’ independent completion of the survey was considered to imply consent to participate in the study” this statement is not clear, kindly clarify

d. Setting: Kindly align your study setting section to fit the actual context of the study according to your study population and objective of your study

e. Please include study design and duration

f. Recruitment and enrollment: How different was the recruitment process between the three different target groups; community members, HCWs and patients? Briefly expand according to your study population

g. Materials: Needs to be specified, Self-administered questionnaire, interviewing forms, online tool? Was it the same method used in each study population?

h. In the materials you mentioned knowledge: Please clarify in the aim of this study if knowledge was also part of it, if yes, please add literature on knowledge as well. Note that since you have different groups, the method to assess knowledge cannot be uniform across

i. Participants and sample: This sample size needs to be justified-what was the basis of 500? You need to justify your sample size and explain how this convenience sampling was done. If your sample size is justified and adequately powered you may wish to stratify your regression analysis by study population

j. Outcome and data analysis “ WHO SAGE Vaccine Hesitancy Survey (see supplementary materials)” should be mentioned in study instruments/materials. Also explanations on each specific question is not necessary. It is also safer to use the term open ended question rather than “qualitative aspect”

4. Results :

a. “We enrolled 516,….508….. however in methods its 500, please be consistent

b. You don’t have to repeat every item found on the table in the text. Please include important items only

c. “When asked if they knew anyone who had ever been hospitalized for COVID-19, 43% responded affirmatively (of them, 45% were healthcare workers); approximately one-third responded that they knew someone who had died due to COVID-19 (of them, 39% were healthcare workers)” . It may not be easy to explain this if healthcare workers, community members and patients are considered under one basket. These groups are different

5. Discussion:

a. In this work, we establish an evidence base for work on COVID-19 vaccine acceptance and hesitancy in Chad through a mixed methods study of patients, community members “ I think this is not a mixed methods study, it is simply a quantitative study with open ended questions”

b. Please clarify the study population in reference 16 and 17

c. One of the strengths mentioned is large sample size, please explain in the method section how this study is powered considering all the three groups. Or modify the statement accordingly

Reviewer #2: This study contributes to the growing body of knowledge in preparation for full COVID-19 campaigns to improve coverage. it highlights key information that need to be addressed through communication to ensure that COVID-19 AND vaccination are understood by the community. However, the tittle says " COVID-19 vaccine acceptance and hesitancy in Tchad. However, they sampled only one reference hospital. Convenience sampling is OK, but to reflect Tchad as a whole, it will be necessary to sample at least a few other hospitals. Moreover, the number of community members was few to represent Tchad. Sampling more patients in other hospitals and participants of other sector of the community will be more representative.

Reviewer #3: This is a great piece on vaccine acceptance and hesitancy in Chad and associated demographic, clinical, and socioeconomic associations.

The authors started with a helpful introduction on the COVID-19 vaccination efforts and challenges in Chad thus far and a rationale as to why it is important to understand vaccine acceptance and hesitancy in order to inform public health campaigns. They also provide a useful background on the Chadian setting and the country’s demographics and existing health and wellness challenges. They then go on to explain how they conducted their survey and collected responses; and then on to highlight the results from their survey and analyses.

While there are some limitations to their methods, the authors clearly acknowledge these and state the rationale for conducting the study in the way that they did. They also discuss how these limitations (e.g. selection bias) may have impacted results and in what direction. Overall, the methods appear to strong despite these issues. Authors stated that anonymous data could be made available on request.

I found the results and discussion to be very well written with the right amount of word space designated to the findings. I found the lack of a statistically significant association between vaccine hesitancy and social media use to be very interesting. Might this have anything to do with the higher levels of education among study participants?

I thought the piece was already very good, and did not have any suggestions for significant changes/improvements (a few very minor edits are listed below). The article is well-written and clearly presented. It will be a great contribution to the literature base as the first account of COVID-19 vaccine acceptance and hesitancy in Chad.

Thank you for the opportunity to review this.

Minor suggestions:

First paragraph of introduction – I suggest rephrasing to something more along the lines of:

Because of testing and reporting issues – particularly early in the pandemic – it is likely that the true number of positive cases in the country is larger than the official count.

Page 2 – “However, there have been no reports” – would suggest adding “To our knowledge” somewhere in this statement

Page 11 – might be useful to define anosmia and ageusia in non-technical terms in parentheses

Page 19, final para – would adjust sentences to “A previous study conducted by the Africa CDC of 15 countries in SSA found XXXXX. While more than two-thirds of participants in our study reported using social media as an avenue for seeking COVID-19 information, we did not find a significant association between higher levels of vaccine hesitancy and higher use of social media, where vaccine mis- and dis-information are likely to prevail.”

In your final paragraph, I suggest adding reference(s) to some of the other barriers of vaccination uptake when you mention this.

6. PLOS authors have the option to publish the peer review history of their article (what does this mean?). If published, this will include your full peer review and any attached files.

**Do you want your identity to be public for this peer review?** For information about this choice, including consent withdrawal, please see our Privacy Policy.

Reviewer #1: No

Reviewer #2: No

Reviewer #3: No

---

## [Decision Letter · Decision Letter 1]

10 May 2022

PGPH-D-22-00016R1

COVID-19 vaccine acceptance and hesitancy in N'Djamena, Chad: A cross-sectional study of patients, community members, and healthcare workers

Dear Dr. Mateen,

Thank you for submitting your manuscript to PLOS Global Public Health. After careful consideration, we feel that it has merit but does not fully meet PLOS Global Public Health’s publication criteria as it currently stands. Therefore, we invite you to submit a revised version of the manuscript that addresses the points raised during the review process.

We look forward to receiving your revised manuscript.

Kind regards,

Joel Msafiri Francis, MD, MS, PhD

Academic Editor

Journal Requirements:

Additional Editor Comments (if provided):

Reviewers' comments:

Reviewer's Responses to Questions

**Comments to the Author**

1. If the authors have adequately addressed your comments raised in a previous round of review and you feel that this manuscript is now acceptable for publication, you may indicate that here to bypass the “Comments to the Author” section, enter your conflict of interest statement in the “Confidential to Editor” section, and submit your "Accept" recommendation.

Reviewer #1: All comments have been addressed

Reviewer #2: All comments have been addressed

Reviewer #3: (No Response)

2. Does this manuscript meet PLOS Global Public Health’s publication criteria? Is the manuscript technically sound, and do the data support the conclusions? The manuscript must describe methodologically and ethically rigorous research with conclusions that are appropriately drawn based on the data presented.

Reviewer #1: Yes

Reviewer #2: Partly

Reviewer #3: Yes

3. Has the statistical analysis been performed appropriately and rigorously?

Reviewer #1: Yes

Reviewer #2: Yes

Reviewer #3: Yes

4. Have the authors made all data underlying the findings in their manuscript fully available (please refer to the Data Availability Statement at the start of the manuscript PDF file)?

Reviewer #1: (No Response)

Reviewer #2: Yes

Reviewer #3: Yes

5. Is the manuscript presented in an intelligible fashion and written in standard English?

Reviewer #1: Yes

Reviewer #2: Yes

Reviewer #3: Yes

6. Review Comments to the Author

Reviewer #1: The authors have responded to the questions asked

Reviewer #2: So much has been published already on COVID-19 vaccine hesitancy in SSA. However, this added to the body of knowledge in Nd'jamena. It offers recommendations on ways to improve vaccine uptake in Tchad which today is still at 1% COVISD-19 vaccination. This may thus be published

Reviewer #3: Dear all,

Thanks for the opportunity to re-review the piece and thank you for considering and addressing the previous reviewer comments. I have very little further to suggest. One small thing I would consider is adding that there are no published reports *to your knowledge* in the introduction section of the abstract.

Additionally, as I was reading through this again, I started to wonder how responses may have differed for people for whom the questions were just hypotheticals versus those who were actually eligible for the vaccination and saw people similar to them (in terms of age, etc.) getting the vaccination. Since the covid-19 vaccine rollout in Chad first began in June 2021 and the survey collection ran from August-October 2021, it’s likely that some of the participants would have been eligible for vaccination when they took the survey, while others weren’t, and this might have played some role in levels of vaccine hesitancy (e.g. older people who see others their age getting the vaccine and not experiencing side effects might feel less hesitant than a young pregnant woman who does not know any people who have similar characteristics to her who have gotten a vaccine). Just another thing to consider, but leave it to you whether it seems like a worthwhile additional discussion point.

Thanks again.

7. PLOS authors have the option to publish the peer review history of their article (what does this mean?). If published, this will include your full peer review and any attached files.

**Do you want your identity to be public for this peer review?** For information about this choice, including consent withdrawal, please see our Privacy Policy.

Reviewer #1: No

Reviewer #2: No

Reviewer #3: No

---

## [Editor Report · Decision Letter 2]

18 May 2022

COVID-19 vaccine acceptance and hesitancy in N'Djamena, Chad: A cross-sectional study of patients, community members, and healthcare workers

PGPH-D-22-00016R2

Dear Dr Mateen,

We are pleased to inform you that your manuscript 'COVID-19 vaccine acceptance and hesitancy in N'Djamena, Chad: A cross-sectional study of patients, community members, and healthcare workers' has been provisionally accepted for publication in PLOS Global Public Health.

Best regards,

Joel Msafiri Francis, MD, MS, PhD

Academic Editor
